# Clinical and Molecular Insights of Arterial and Venous Thrombosis in Myeloproliferative Diseases—Case-Based Narrative Review

**DOI:** 10.3390/biomedicines13102543

**Published:** 2025-10-18

**Authors:** Anca Drăgan, Mădălina Găvănescu, Adrian Ştefan Drăgan, Alexandru Bardaş, Monica Dobrovie, Anca Doina Mateescu

**Affiliations:** 1Department of Cardiovascular Anaesthesiology and Intensive Care, “Prof. Dr. C.C. Iliescu” Emergency Institute for Cardiovascular Diseases, 258 Fundeni Road, 022328 Bucharest, Romania; 2Department of Vascular Surgery, “Prof. Dr. C.C. Iliescu” Emergency Institute for Cardiovascular Diseases, 258 Fundeni Road, 022328 Bucharest, Romania; madalinagavanescu@yahoo.com; 3Faculty of General Medicine, Carol Davila University of Medicine and Pharmacy, 8 Eroii Sanitari Boulevard, 050474 Bucharest, Romania; ancad.mateescu@gmail.com; 4Department of Hematology and Bone Marrow Transplant, Fundeni Clinical Institute, 258 Fundeni Road, 022328 Bucharest, Romania; hematologie2@icfundeni.ro; 5Department of Radiology, Fundeni Clinical Institute, 258 Fundeni Road, 022328 Bucharest, Romania; monicadobrovie@gmail.com; 6Department of Cardiology, “Prof. Dr. C.C. Iliescu” Emergency Institute for Cardiovascular Diseases, 258 Fundeni Road, 022328 Bucharest, Romania

**Keywords:** arterial thrombosis, aortic thrombi, portal thrombi, multidisciplinary approach, myeloproliferative neoplasm, vascular, acute ischemia

## Abstract

The myeloproliferative neoplasms (MPN), a heterogeneous group of disorders characterized by specific genetic mutations, have the development of arterial and venous thrombosis as their main complication. Almost 40–50% of MPN patients encountered arterial or venous thrombosis during the course of their disease. Moreover, arterial thrombosis is linked to significant mortality, progression to myelofibrosis, and an increased risk of developing second cancers. Despite significant advancements in medical research, there are still unmet needs in this field. Our narrative review provides clinical and genetic insights into thrombosis associated with myeloproliferative neoplasms. We focus on the underlying pathophysiological processes, assessment methods, and risk stratification related to thrombotic events. This information aims to assist clinicians in accurately assessing the risks associated with MPN thrombosis, enabling a more personalized and effective approach to patient care. We based our review on a rare case of MPN-associated thrombosis, whose clinical presentation was marked by acute ischemia in both lower limbs. The thrombosis affected the distal aortic arch, thoracic and abdominal aorta, celiac trunk, common and proper hepatic arteries, proximal left renal artery, several segmental arteries in the right kidney, and the portal vein thrombosis. Our review presents various therapeutic options for these conditions. In the presented case, the multiple thrombi were treated medically, except for the popliteal artery thromboses, which required surgical management. This case may serve as a valuable reference for choosing treatment options for aortic and portal vein thrombosis, highlighting the multidisciplinary approach.

## 1. Introduction

*BCR-ABL1*-negative myeloproliferative neoplasms (MPN) represent a heterogeneous group of disorders characterized by specific genetic mutations that result in the abnormal clonal expansion of the hematopoietic cell proliferation, leading to the overproduction of leukocytes, erythrocytes, or platelets. The main MPN types include polycythemia vera (PV), essential thrombocythemia (ET), primary myelofibrosis (PMF), and unclassifiable MPN (MPN-U) [1,2].

The MPN complications are related to the development of bone marrow fibrosis, leukemic transformation, and arterial and venous thrombosis [3]. Zhang et al. reported that 40–50% of patients with MPN encountered arterial or venous thrombosis during the course of their disease [4]. Notably, arterial thrombosis is linked to significant mortality [5,6,7,8,9], progression to myelofibrosis [5,6,7], and an increased risk of developing second cancers [5,10,11,12]. Thrombosis may arise prior to, at the time of, or subsequent to the diagnosis of MPN. Hultcrantz et al. demonstrated that the rate of thrombosis was significantly higher in MPN patients compared to the general population, with the highest hazard ratio observed shortly after the MPN diagnosis [13]. It is well known that thrombosis in MPN occurs in the medium-sized arteries and veins, and also in the microvessels [14]. Unusual sites of venous thrombosis include upper extremity veins, splanchnic veins, cerebral veins, and retinal veins, and unusual sites of arterial thrombosis include renal, adrenal, splenic and mesenteric arteries, and intracardiac and aortal locations [15]. Despite remarkable advancements in medical research, the persistent prevalence of arterial and venous thrombosis remains a serious challenge in the healthcare landscape. Current efforts to assess and stratify thrombotic risk have progressed from conventional methods to sophisticated risk scores and machine learning algorithms, all designed to tailor patient management more effectively. Moreover, gaining a deeper understanding of the pathophysiology of thrombotic events in the MPN context is crucial; this knowledge not only enhances risk assessment but also paves the way for targeted and more effective treatments. Furthermore, it is imperative to raise awareness of these serious manifestations and embrace a multidisciplinary approach to their management. By doing so, we can improve outcomes and enhance the quality of care for patients facing these conditions.

Our narrative review offers clinical and genetic insights into thrombosis associated with myeloproliferative neoplasms. We focus on pathophysiological processes, assessment methods, and risk stratification related to thrombotic events. This information aims to assist clinicians in accurately assessing the risks associated with MPN thrombosis, enabling a more personalized and effective approach to patient care. To illustrate the complexities of this condition, we present a rare case of MPN-U associated thrombosis affecting the distal aortic arch, thoracic and abdominal aorta, celiac trunk, common and proper hepatic arteries, proximal left renal artery, as well as several segmental arteries in the right kidney. Additionally, we report portal vein thrombosis (PVT) in this case. The clinical presentation was marked by acute ischemia in both lower limbs, highlighting the need for a multidisciplinary approach to management. Previously reported cases of MPN thrombosis in large vessels, such as the aorta and iliac artery, have typically required surgical intervention [16]. However, urgent surgical treatment of aortic thrombi is associated with a high mortality rate. There are only a few documented instances of medical management for an MPN-associated aortic thrombus [17]. Similarly, PVT is rare and can sometimes be the first clinical indication of an underlying MPN [18]. Our review presents various therapeutic options for these conditions. In our case, multiple thrombi were treated medically, except for the popliteal artery thromboses, which required different management. This case may serve as a valuable reference for choosing treatment options for aortic and portal vein thrombosis.

## 2. Case Presentation

A 64-year-old woman was referred to our hospital with acute bilateral lower limb ischemia. Our patient complained of sudden coldness, paresthesia, and pain in the lower limbs that began 8 h before the presentation to the emergency department. Apparently, the patient was in good health, without a significant medical history, except for a diagnosis of uninvestigated thrombocytosis and leukocytosis for almost one year. Moreover, she described diffuse abdominal pain for the past three months, which she had ignored. The patient denied fever, cough, headache, melena, and tarry stool. She had no chest pain, syncope, or palpitations. There was no family history of coagulation disorders. She denied a personal or family history of cerebrovascular attack and ischemic heart disease. The patient had no associated cardiovascular risk factors and was not taking any medications at home.

Examination in the emergency room showed the abolition of both the right and left pedal and posterior tibial arterial pulses. The other peripheral pulses were present. Both lower limbs were pale and cold, with mild loss of sensibility limited to the left foot. Normal motor function was evidenced. Otherwise, her physical examination revealed no other abnormal clinical signs or symptoms, except for an enlarged spleen. Vital signs were within normal range: a blood pressure of 140/64 mmHg, a regular pulse rate of 76/min, and a respiratory rate of 16 per/minute. Her body temperature was 36.7 °C. At admission, our patient had a leukocytosisof 31 × 10^3^/μL with 93% neutrophils and 3% lymphocytes. a hemoglobin level of 16 g/dL, and platelet count of 621 × 10^3^/μL. A mild increase in liver enzymes was also present (AST/ALT of 167/204 UI/L). No other abnormalities were detected in other standard laboratory results. No proteinuria nor hematuria was found. Doppler ultrasound examination confirmed the occlusion of the arterial trunks in both lower limbs. Cardiological evaluation in the emergency department revealed a normal electrocardiogram with no cardiac abnormalities at transthoracic echocardiography.

Emergency surgery was necessary with a diagnosis of acute bilateral lower limb ischemia. Under general anesthesia, bilateral popliteal and tibial arterial axis dethrombosis with restoration of arterial flow was performed via the distal popliteal route with removal of multiple thrombi. However, the etiology of the acute lower limb ischemia remained uncertain in a patient in sinus rhythm. Moreover, the transesophageal echocardiography (TOE) performed in the operating room revealed multiple thrombi of various forms, highly mobile, attached to the aortic arch and the wall of the entire descending thoracic aorta. (Figure 1). Doppler ultrasound of the extracranial carotid arteries did not reveal any abnormalities.

Due to TOE findings, we performed, early postoperatively, a computed tomography (CT) scan of her thorax and abdomen, which showed the presence of multiple thrombi in the distal aortic arch, thoracic, and abdominal aorta, celiac trunk, common and proper hepatic arteries, proximal left renal artery, and some segmental arteries at the level of the right kidney. Arteriosclerosis was not recognized (Figure 2 and Figure 3).

Moreover, there was a completely obstructive thrombosis of the left portal vein, extended to the portal vein trunk and the splenic vein, while the superior mesenteric vein and the iliac veins remained permeable (Figure 4).

Vascular-ischemic changes were noted in the spleen (almost total damage), left kidney (extensive ischemic changes in the upper pole, in the middle third, and partial changes in the lower pole), right kidney (partial ischemic changes in the upper pole and in one-third), and the left hepatic lobe. There was a fluid accumulation adjacent to the pericardium on the left side, measuring 97/57 mm, most likely due to left incised pleurisy. The liver exhibited ischemic changes in the left lobe, accompanied by perfusion disorders in the pancreas and adrenal glands. No signs of oncological disease were detected at the CT scan.

At admission to the intensive care unit (ICU), the parenteral anticoagulant therapy with unfractioned heparin, along with aspirin as antiplatelet treatment, was started. The postoperative course was favorable and uneventful, with the patient being fast extubated in the ICU. Hydroxyurea was initiated on the second day of her admission and uptitrated to a maximum dose of 3g/day (Figure 5).

On the 11th day of admission, the dose of hydroxyurea was reduced to 2 g/day. The numbness in both of her lower extremities disappeared within two days. Additionally, the number and dimensions of the aortic thrombi decreased at the repeated TOE and CT-scan on the 7th day of her hospital stay (Figure 6).

The hematological evaluation raised the suspicion of a myeloproliferative disorder. The bone marrow aspirate revealed a hypercellular marrow and panmyelosis, an appearance compatible with a chronic MPN-U. Moreover, the JAK2 V617F mutation burden was 13%. The hematologist diagnosed this case as MPN-U JAK2 V617F positive, and the aortic thrombus was thought to be associated with MPN. An MPN-U is a hematological neoplasm characterized by clonal proliferation of myeloid precursors in the bone marrow, blood, and other tissues (spleen, liver); however, it does not meet the specific diagnostic criteria for a distinct type of MPN. It is a diagnosis of exclusion used when clinical, morphological, and molecular features of an MPN are present, but insufficient for a definitive classification into categories like PV or essential thrombocythemia. On one hand, it is characterized by dysplasia and ineffective hematopoiesis, and on the other hand, it is characterized by an increase in the number of more than one type of myeloid lineage. Symptoms can be non-specific and variable, including fatigue, weight loss, and splenomegaly and hepatomegaly, particularly in later stages.

The patient was transferred to the Hematological Department, where she continued medical treatment with enoxaparin 1 mg/kg twice-daily initially, aspirin 75 mg/day, and hydroxyurea 1.5 g/day. At discharge, she received novel anticoagulant therapy with apixaban 5 mg bi-daily along with aspirin 75 mg/day and hydroxyurea 1 g/day for 5 days/week and 1.5g/day for 2 days/week. At the 3-month CT follow-up (Figure 6), we noted the disappearance of the aortic thrombus. However, CT revealed persistent stenosis at the origin of the celiac trunk, persistent left PVT with small adjacent collaterals, and post-thrombotic occlusion of the splenic vein with collateral formation and necrosis of about 80% of the splenic volume. The patient presented no symptoms at 6-month follow-up.

## 3. Insights into Pathophysiological Mechanisms of Thrombosis in Myeloproliferative Neoplasms

The mechanism of thrombosis in MPN is complex and multifactorial. Various components, including the quantity and activation patterns of blood cells, endothelial involvement, coagulation disorders, and thrombo-inflammatory processes, all play significant roles in this phenomenon (Figure 7).

### 3.1. Genetic Mutations

Notably, genetic mutations exert a substantial influence on the overall environment, thereby impacting thrombosis development. JAK2/CALR/MPL gene mutations are the driver mutations in MPN [19,20]. One of the most common mutations in the Janus kinase 2 (JAK2) gene is the V617F, which is a somatic gain-of-function mutation. This mutation alters the 1849th coding nucleotide, changing it from guanine (G) to thymine (T), resulting in the replacement of the amino acid valine with phenylalanine [20]. JAK2 encodes a non-receptor tyrosine kinase involved in cell growth, differentiation, development, or histone modification. JAK2 facilitates the phosphorylation of STAT proteins, which, upon phosphorylation, dimerize in the cytoplasm and are transported into the nucleus to activate specific genes. Among these mutations, JAK2V617F was the most prevalent driver mutation in MPN associated with thrombotic complications. Moreover, thrombosis may be exacerbated by co-occurring mutations such as TET2, DNMT3A, and ASXL1 [19,20].

Research suggested that even in populations lacking a formal MPN diagnosis, individuals with a JAK2V617F allele burden above 1% faced a heightened risk of venous thromboembolism compared to those without the mutation [21]. Additionally, Cordua et al. reported a JAK2 V617F prevalence of 3.1% in their study of the Danish general population [21]. In MPN diseases, Vignoli et al. demonstrated that patients with more than 50% JAK2V617F allele burden had a significantly higher platelet adhesion capacity compared to subjects with less than 50% JAK2V617F allele burden, thus promoting thrombosis [22].

A single mutation in the JAK2 gene cannot fully explain the diverse MPN landscape. While no link has been established between MPL mutations and thrombosis, a significant correlation has been reported between the JAK2 V617F mutation and ASXL1 gene mutations in relation to thrombotic events [20]. However, the underlying mechanism related to the ASXL1 mutation remains incompletely understood [20]. Furthermore, mutations in the TET2 gene have been identified in approximately 15% of MPN patients [23]. TET2 plays various roles during hematopoiesis, including stem cell self-renewal, lineage commitment, and the terminal differentiation of monocytes. Additionally, TET2 facilitates JAK2 or STAT5 signal transduction and modulates the epigenetic composition of genomic DNA, influencing both DNA and histone methylation and acetylation [23]. The order in which mutations occur influences the proliferative response to JAK2 V617F and affects the ability of double-mutant hematopoietic and progenitor cells to form colonies. In patients where TET2 mutations occurred first, the hematopoietic stem and progenitor cell compartment was primarily made up of TET2 single-mutant cells. In contrast, in patients where JAK2 mutations occurred first, the compartment was dominated by JAK2-TET2 double-mutant cells. Prior mutation of TET2 altered the transcriptional effects of JAK2 V617F in a cell-intrinsic manner, preventing JAK2 V617F from upregulating genes associated with proliferation [24]. DNMT3A mutations, an independent risk factor for arterial thrombosis in MPN, leads to aggressive inflammatory transcriptome, enhance macrophage activity and stimulate proinflammatory T-cell polarization [25]. Morath et al. suggested a similar relationship for DNMT3A with JAK2 V617F as TET2 in thrombotic events [25]. MPN thrombosis is also related to epigenetic dysregulations, that refers to the mutations in genes that encode proteins which regulate the chromatin structure (ASXL1, TET2, IDH1/2, EZH2, IKZF1, JAK2 V617F, and PRMT5), and to the methylation status of promoter sites of genes that coordinate cell growth, differentiation, and survival [20].

### 3.2. The Coagulation Disorders and Platelet Involvement

Virchow’s triad described the conditions that lead to thrombosis endothelial injury/dysfunction, stasis of blood flow, and hypercoagulability. Studies have shown that MPN patients exhibit a continuous hypercoagulable state, even in the absence of overt thrombosis [26]. Levels of proteins C and S were significantly lower in MPN patients compared to control subjects [26]. Whole blood rotational thromboelastometry (ROTEM) revealed a hypercoagulable profile in MPN patients [27]. Giaccherini et al. showed that the ROTEM parameters were significantly affected by platelet count. After adjusting for platelet count, the maximum clot firmness (MCF) values indicated reduced platelet reactivity in MPN patients, supporting the hypothesis that platelet function becomes exhausted during clotting activation [27]. Higher mean MCF values were also found in the ET and PV settings by Şahin et al. [28]. Additionally, the PDW and (mean platelet volume) MPV values suggested the presence of large-sized platelets, with a higher variation in size, especially in ET patients [26]. Moreover, the increase in thrombin generation may be attributed to platelets that arise from the clonal process, resulting in ongoing activation of the TGF-beta/ALK5 and PAR-2/PAR-1 signaling pathways [29].

Recently, Ross et al. demonstrated that there may be a thrombotic risk associated with increased mitochondrial mass in platelets from MPN patients, although no correlation between the variant allele frequency (VAF) of JAK2 V617F (measured in leukocyte DNA) and mitochondrial mass in platelets was found [30]. They found that the mitochondrial membrane potential is altered in circulating platelets before stimulation, and that MPN platelets are more sensitive to mitochondrial membrane depolarization, leading to the formation of balloon-shaped platelets. Additionally, mitochondria in MPN platelets may have a greater tendency to be released as microparticles [30].

### 3.3. The Leukocytes Involvement

Platelet activation has also the capacity to stimulate neutrophils [19]. Vignoli et al. reported that platelet P-selectin binding to PSGL-1 activates neutrophils, leading to increased surface CD11b expression, cathepsin-G release from neutrophil granules, and the generation of procoagulant microparticles from both platelets and leukocytes [22].

Nayak et al. highlighted the role of activated neutrophils in both arterial and venous thrombosis, identifying the transcription factor Krüppel-like factor 2 (KLF2) as a key regulator of this process [31]. Activated neutrophils cluster P-selectin glycoprotein ligand 1 (PSGL-1) through cortical actin remodeling, increasing their adhesion at thrombosis sites. This pathway may be effectively targeted using immunoregulatory nanoparticles [31]. Zhang et al. reported that neutrophil extracellular traps (NETs) was significantly enriched in the neutrophils of PV patients, leading to thrombosis through the PRKCD-mediated NETs pathway [32].

The expression of multiple genes associated with hypoxia-inducible factor (HIF) might increase due to the JAK2-V617F mutation in leukocytes. HIF activates pathways associated with inflammation and thrombosis by modulating the production of KLF2 and vascular endothelial growth factor (VEGF) [19]. Additionally, the activated leukocytes had elevated phosphatidylserine expression enhancing the formation of tenase and prothrombinase complexes. This process subsequently increases the production of FXa, thrombin, and fibrin, thereby shortening coagulation time [19]. Moreover, the neutrophil-derived microparticles expose active CD11b/CD18 integrin molecules that lead to platelet activation and P-selectin expression [33].

### 3.4. The Thrombo-Inflammatory Process

Leukocytes are crucial in the thrombo-inflammatory MPN process. They release various proinflammatory and prothrombotic factors and interact with other cells, megakaryocytes, and ECs, contributing to the amplification of the clotting cascade and subsequent thrombosis [19]. Elevated platelet-leukocyte aggregates were reported to be an independent risk factor for MPN thrombosis, indicating that platelets could be a mediator between thrombosis and inflammation [34,35]. He et al. identified the PI3K/AKT/mTOR signaling pathway as a significant contributor to platelet dysregulation, and reported that metabolic intervention via α-KG supplementation suppressed platelet activation by inhibiting this pathway, thereby reducing hyperactivation associated with MPN [35].

Platelets also exhibit immune cell-like behavior, leading to the release of various inflammatory mediators, thereby promoting inflammation-based thrombosis [19]. Thrombo-inflammation is the phenomenon involving activation of Toll-like receptor 2 on platelets, which leads to inflammation and promotes thrombosis. Later, this was defined as a pathological response occurring in the vasculature following blood vessel injury [19]. The inflammatory state leads to the production of microparticles that damage vascular endothelial cells (ECs). High levels of these circulating microparticles, which contain prothrombotic mediators, alter the inflammatory response, activate blood cells, and disrupt the coagulation system [33].

### 3.5. The Endothelial Involvement

JAK2V617F mutation can alter endothelial function. The JAK-STAT pathway leads to vascular activation, while the JAK-STAT inhibitors reduce critical proadhesive and prothrombotic markers, and prevent increased leukocyte-endothelial adhesion [36]. ECs JAK2V617F-mutated promote thrombosis through induction of endothelial P-selectin expression [36,37], and von Willebrand factor (VWF) [36]. Furthermore, Diz-Küçükkaya et al. reported for the first time that the JAK2V617F mutation was identified somatically in both peripheral blood cells and ECs from patients with atherosclerosis [38]. The authors suggested that this finding may support the idea that ECs and hematopoietic cells originate from the same clone or that somatic mutations could be transmitted to ECs through other mechanisms that require further investigation [38]. Previously, Hekimoğlu et al. were the first to demonstrate the significance of endothelial microparticles released by ECs carrying the JAK2V617F mutation [39]. These microparticles play a crucial role in cellular communication, triggering inflammation and thrombosis. Their production should be viewed as a result of cellular activation initiated by the JAK2V617F mutation, rather than merely a sign of endothelial cell dysfunction [39].

Šefer et al. conducted a prospective analysis to examine the impact of activated blood cells and ECs on thrombosis in MPN patients. They investigated the correlation between circulating leukocyte-platelet aggregates and soluble selectins. Their results revealed that monocyte-platelet aggregates are an independent risk factor for thrombosis in this patient population [34].

### 3.6. Red Blood Cells Involvement

An elevated red blood cell (RBC) count influences the thrombotic process by increasing blood viscosity and slowing blood flow, thereby serving as a strong prothrombotic factor. Even a minor reduction in RBC deformability can significantly increase microvascular flow resistance and blood viscosity, potentially resulting in thrombotic complications [40,41]. Blood viscosity is a recognized risk factor for thrombosis and increases nonlinearly with hematocrit [39,40]. RBCs promote arterial thrombosis by increasing thrombus growth through a platelet-dependent mechanism that does not rely on thrombin generation [41]. Therefore, hematocrit has a significantly greater impact on platelet adhesion than the platelet count [22]. An increased hematocrit promotes platelet margination, permitting shear-induced platelet aggregation through αIIbβIII-mediated adhesion at supraphysiological shear rates [42].

Recent studies have identified erythrocyte-derived microvesicles that are associated with a dose- and time-dependent increase in thrombin generation and a reduction in clotting time, indicating an enhancement of hypercoagulability [43]. RBCs in MPN patients exhibit greater adhesion to the endothelium as a result of elevated levels of Lutheran/BCAM protein (Lu) on their surface [44,45]. The overexpression of Lu/BCAM facilitates binding to laminin α5, which is expressed on ECs. Researchers proposed that variability in the expression of adhesion molecules throughout the vascular system, as well as differences in hemodynamic conditions, may account for the site-specific occurrence of thrombosis [44].

## 4. Thrombosis Risk Assessment and Stratification

### 4.1. Clinical Assessment

Age has been identified as a risk factor for arterial thrombosis, but not for venous thrombosis in MPN patients [46,47,48]. Barbui et al. included for the first-time patients aged more than 60 years old and those with a history of thrombosis in their assessment of high-risk patients regarding thrombotic events in MPN [49]. Individuals aged 60 years and older have been integrated into the newly established risk scores designed to predict thrombosis. In the European Collaboration on Low-Dose Aspirin in Polycythemia Vera (ECLAP) prospective multicenter project, age older than 65 years and prior thrombotic events were identified as risk factors for cardiovascular events among PV patients [50,51]. The older age, the use of JAK inhibitors or low-molecular-weight heparin, previous arterial or venous events, were identified as independent risk factors for new arterial or venous events in a recently published analysis of 1079 MF patients [52]. Advanced age and a prior history of thrombotic events have been shown to be significantly associated with the occurrence of thrombosis in adult PV patients according to the findings of the REVEAL study [53]. Ripamonti et al. demonstrated that age above 60 years old predicted independently only the venous thrombosis, not the arterial events [54]. However, a study by Shimano et al. observed a 12% rate of thrombosis even in a pediatric cohort. In this group, there was a significantly higher incidence of thrombosis in patients with JAK2V617F mutations and those diagnosed with PV [55]. Wille et al. showed that more than half of the thromboembolic complications (53.89%) occurred before or at the time of MPN diagnosis [56].

Researchers reported sex differences regarding thrombosis associated with MPN. Women show unique patterns of thrombosis and conflicting bleeding tendencies [57]. Notably, they have a higher prevalence of unusual thromboses, such as cerebral and splanchnic vein thrombosis. Additionally, women are at an increased risk of hemorrhage due to anti-thrombotic medication, as well as conditions like acquired von Willebrand syndrome and platelet dysfunction [57]. However, female sex was associated with more favorable clinical outcomes, including lower rates of progression to myelofibrosis or leukemia, as well as a reduced incidence of secondary cancers [58]. Venous thrombosis was correlated with female sex, regardless of age, phenotype at diagnosis, or MPN-specific mutations [53,59]. In PV, females with palpable splenomegaly and a history of major hemorrhage were more likely to exhibit a venous thrombotic event [60]. Specifically, Karantanos et al. reported a venous thrombosis incidence of 14.7% in women compared to 7.2% in men [59]. The REVEAL study also indicated that male sex was significantly linked to a reduction in thrombosis events, while female sex was risk factor [53]. In contrast, studies reported no difference in the incidence of arterial ischemic events between sexes [53,59]. Furthermore, Shimano et al. found no difference in thrombosis incidence between sexes, studying a pediatric cohort [55].

An important issue to consider in MPN arterial thrombosis is the increased risk of secondary cancer (SSC). Hindilerden et al. recently reported a bidirectional link between arterial thrombosis in MPN patients and SSC in a retrospective multicenter study [12]. They also emphasized the protective effect of interferon-based therapy against SSC and the need for SCC search in MPN patients with arterial thrombosis. Carcinoma emerged as the most prevalent form of SSC among MPN patients, with older age at the time of MPN diagnosis and male sex identified as significant risk factors [12].

### 4.2. Cardiovascular Risk Factors

The cardiovascular risk factors have been shown to influence the incidence of thrombosis in MPN patients. Gu et al. identified independent risk factors for thrombosis in PV patients, including age 60 or older, cardiovascular risk factors, at least one high-risk mutation (DNMT3A, ASXL1, or BCOR/BCORL1), and a history of previous thrombosis [61]. Obesity, dyslipidemia, diabetes, hypertension, and smoking predicted the thrombotic complications after MPN diagnosis in the same cohort (HR 4.22; CI 95%: 2.00–8.92) [61]. Aswad et al. showed that a history of thrombosis (HR 2.23; *p* 0.019), age over 60 years at diagnosis (HR 1.56, *p* 0.037), the presence of JAK2V617F mutation (HR 3.01, *p* 0.002), and tobacco smoking (HR 1.75, *p* 0.01) were independent risk factors of the thrombotic complications during the follow-up [62]. Notably, arterial hypertension has been identified as a significant risk factor for both thrombosis and kidney dysfunction in the MPN population in Gecht et al. study [63]. On the contrary, Ripamonti et al. demonstrated in a ET cohort that diabetes mellitus (DM) and hypertension predicted only the arterial thrombotic events [54]. Recently, How et al. conducted the largest retrospective analysis assessing the impact of cardiovascular (CV) risk factors in MPN patients within a real-world population. Their findings indicated that the risk of arterial thrombosis was particularly pronounced in PV patients. Specifically, having one or more CV risk factors was associated with nearly a six-fold increase in the likelihood of subsequent arterial thromboses [64]. Additionally, MF patients were more likely to have a CV risk factor compared to those with ET and PV, especially concerning hypertension and hyperlipidemia. Furthermore, DM2 emerged as an independent risk factor for both arterial and venous thrombosis in PV (HR 3.60; CI 95%: 1.11–11.68) and MF (HR 4.09; CI 95%: 1.20–13.95), but not in ET (HR 0.90; CI 95%: 0.19–4.34) [64]. Previously, Cerquozzi et al. found diabetes to be associated with arterial thrombosis, along with older age (≥60 years), hypertension, diabetes, hyperlipidemia, and normal karyotype [60]. Recently, Krecak et al. established a prognostic cutoff for HbA1c at ≥7.2% to assess the thrombotic risk in MPN patients [65]. Their research found no correlations between HbA1c levels and various blood cell count components at the time of diagnosis. However, they identified significant interactions between high HbA1c levels, older age, and different mature myeloid cell lineages, indicating that uncontrolled HbA1c may act synergistically with the MPN clone, contributing to cardiovascular complications in MPN patients [63]. Hyperlipidemia (HR 1.8; CI 95%: 1.1–2.9) and prior arterial events (HR 2.7; CI 95%: 1.7–4.5) emerged as predictors of subsequent arterial events in the same PV cohort [58]. In a different study, Furuya et al. discovered that hypertriglyceridemia (HR 3.364; CI 95%: 1.541–7.346) acted as an independent risk factor for thrombosis in a Japanese population diagnosed with ET [66]. They also noted that patients with serum triglyceride levels of ≥1.2 mmol/L or those with two or more cardiovascular risk factors tended to have poorer thrombosis-free survival [66]. Additionally, Zhang et al. reported that dyslipidemia was more prevalent among MPN individuals who experienced thrombotic events compared to those who remained thrombosis-free (62.1% vs. 37.9%; *p* < 0.0001) [67]. The Lai et al.’ study demonstrated for the first time that inflammation and abnormal lipid metabolism were closely related to the high risk of thrombosis in PV patients [68]. Lipid abnormalities contribute to the development of atherosclerosis and interact with oxidative stress, inflammation, genetics, and various other factors to increase the risk of thrombosis in MPNs [69]. Recently, Enblom-Larsson et al. found that previous ischemic heart disease and hypertension were significantly associated with an increased risk of a new arterial event in both PV and ET [52]. COVID-19 was also reported in MPN patients as a risk factor for arterial thrombosis but not VTE, bleeding, and death compared with non-MPN patients [70].

In conclusion, it is imperative for clinicians to recognize that cardiovascular risk factors, which predominantly align with those associated with atherosclerosis, must be effectively managed in MPN patients to mitigate the risk of thrombosis events.

### 4.3. Blood Cells Assessment

Researchers investigated blood cells in relation to the risk of thrombosis in MPN patients. In their prospective, interventional MAJIC-PV study, Harrison et al. discovered that achieving normalization of all blood counts results in significantly longer thrombosis event-free survival compared to patients who did not reach this goal [71]. In PV setting, maintaining hematocrit levels below 45% was recommended to reduce the thrombotic risk [72,73]. Furthermore, the Cytoreductive Therapy in Polycythemia Vera (CYTO-PV) large-scale, multicenter, prospective, randomized clinical trial reported that maintaining a hematocrit target of 45% to 50% through conventional treatments (including phlebotomy, hydroxyurea, or a combination of both) resulted in a fourfold increase in the rate of death from cardiovascular causes or major thrombosis, compared to maintaining a hematocrit target of less than 45% [74]. Chojecki et al. showed that ruxolitinib was effective in controlling hematocrit levels after three and six months of treatment in PV patients, leading to a lower thrombotic risk [75]. In contrast, Shimoda et al. found in a Japanese cohort that leukocyte counts exceeding 15 × 10^9^/L and platelet counts above 1000 × 10^9^/L were independent predictors of thrombosis in PV patients, rather than hematocrit levels above 45% [76].

A recent analysis from the REVEAL study provided valuable insights into the relationship between hematocrit levels, leukocyte counts, platelet counts, and absolute neutrophil counts and the risk of thrombosis in PV patients. This research is the largest prospective multicenter observational study in this setting. A hematocrit level higher than 45%, a leukocyte count exceeding 11 × 10^9^/L, a platelet count above 400 × 10^9^/L, and an absolute neutrophil count greater than 7 × 10^9^/L are all associated with the occurrence of thrombosis in adult PV patients [53]. Specifically, a hematocrit level above 45%, a platelet count over 400 × 10^9^/L, and older age significantly correlated with an increased risk of arterial thrombosis [53]. Interestingly, a leukocyte count greater than 11 × 10^9^/L was significantly linked to both venous and arterial thrombosis. Even when hematocrit levels were below 45%, a leukocyte count exceeding 11 × 10^9^/L was linked to an approximately twofold increase in thrombotic risk compared to patients with leukocyte counts below this threshold [53]. These findings emphasized the importance of monitoring leukocyte levels in PV patients to evaluate and manage their thrombosis risk effectively. Cerquozzi et al. previously found that the presence of major hemorrhage at diagnosis, a leukocyte count above 11 × 10^9^/L, and history of prior venous events predicted subsequent venous thrombotic events, not arterial ones, in patients with PV [60].

Carobbio et al. demonstrated that the influence of leukocytosis was more pronounced in ET (RR 1.65; CI 95%: 1.43–1.91) compared with PV patients (RR 1.34; CI 95%: 1.08–1.66). This effect appears to be exclusively associated with arterial events (RR 1.45; CI 95%: 1.13–1.86) because no effect was found in venous setting (RR 1.14; CI 95%: 0.65–1.98) [77]. In a previous analysis by Carobbio et al., several independent risk factors for arterial thrombotic events in ET patients were identified, including age over 60 years, a history of thrombosis, the presence of cardiovascular risk factors (such as tobacco use, hypertension, and DM), a leukocyte count higher than 11 × 10^9^/L, and the presence of the JAK2V617F mutation [46].

Regarding thrombocytosis, Galvez et al. reported that many retrospective and prospective trials argued against associations between thrombocytosis and risk of thrombosis in patients with ET and PV. Furthermore, most studies indicate that extreme thrombocytosis was linked to a higher risk of hemorrhagic events, a paradoxical situation with significant clinical implications [78]. In the JSH-MPN-R18 study, thrombocytosis was identified as a risk factor for hemorrhagic events and prognosis but not for thrombotic events [79]. Conversely, the Reveal trial found that a platelet count exceeding 400 × 10^9^/L was associated with the occurrence of thrombosis in adult patients with PV [53]. Shimoda et al. also reported that platelet counts above 1000 × 10^9^/L one of the independent predictors of thrombosis in PV patients [76]. In a study by Lucijanic et al., the platelet count also exhibited significant results in correlation with a shorter time to thrombosis in prefibrotic patients (values greater than 752 × 10^9^/L) and in overt fibrotic patients (values lower than 385 × 10^9^/L) [80]. The thrombotic risk in MPN patients was not directly correlated with platelet count, but rather with the factors that mediate aberrant platelet activation in this setting [35].

Other indexes could help risk assessment in MPN thrombosis. The SHIP-study described red blood cell distribution width (RDW) as an independent predictive parameter for thromboembolic events in the general population [81]. Their research demonstrated a significant association between an early high RDW and the diagnosis of acute unprovoked venous thrombosis. Specifically, a RDW of 14% or greater was identified as an independent predictor of unprovoked venous thrombosis in adult patients [82]. Higher RDW means decreased RBC deformability leading to erythrocyte aggregation, increased blood viscosity, and thrombotic susceptibility [43]. Furthermore, several studies have shown that RDW can predict thrombosis in patients with PV [43,83] and ET [43,80]. The absolute neutrophil count has been identified as an independent risk factor for thrombotic events in PV patients before or at the time of diagnosis, according to Zhang et al. [32]. In a study by Lucijanic at al., the absolute neutrophil count higher than 8.33 × 10^9^/L, and, respectively, 8.8 × 10^9^/L was also significantly associated with a shorter time to thrombosis in prefibrotic patients and overt fibrotic patients [80]. Absolute lymphocytes count also exhibited significant results in correlation with a shorter time to thrombosis in prefibrotic patients (values greater than 2.58 × 10^9^/L) and in overt fibrotic patients (values lower than 1.43 × 10^9^/L) [80]. In Carobbio et al.’s research, the history of previous venous events (HR = 5.48, *p* ≤ 0.001), and the neutrophil-to lymphocyte ratio (NLR) greater than 5 (HR = 2.13, *p* = 0.001) were strongly associated with the risk of venous thrombosis in PV patients [84]. Furthermore, Lai et al. found the neutrophil/HDL ratio (NHR) was the most effective predictor of thrombosis in PV patients, outperforming the NLR in this setting (AUC = 0.791 compared to 0.658) [68]. Regarding ET, Ripamonti et al. demonstrated that an NLR higher than 4 predicted arterial thrombotic events, along with several cardiovascular risk factors (DM and hypertension) [54]. Additionally, they found that individuals over 60 years old and an NLR greater than 5 independently predicted venous thrombotic events [54].

In conclusion, clinicians should be cautious to maintain hematocrit levels below 45% and avoid leukocytosis to prevent thrombosis events. Additionally, monitoring platelet count and function is essential, as these factors may predict both thrombosis and hemorrhage.

### 4.4. Genetic Risk Factors

The primary genetic mutations identified in MPN patients are JAK2, CALR, and MPL. A strong correlation was reported between thrombotic events and mutations in the JAK2 and ASXL1 genes. In contrast, MPL mutations have not been associated with thrombosis [20]. Additionally, mutations in epigenetic modifier genes such as TET2, ASXL1, DNMT3A, and EZH2 occur frequently in MPN cases, with reported frequencies ranging from 1% to 30% [20].

In a study by Ha et al., the allele JAK2V617F burden was found to be highest in PV (66.0% ± 24.9%) compared with ET (40.5% ± 25.2%) and PMF patients (31.5% ± 37.0%) [85]. In contrast, a study by Chatambudza et al. involving an African cohort reported a significantly lower allele burden in ET patients (24.9%), but higher allele burden in PV (71.1%) and PMF (55.8%) patients [86]. In PV patients, a VAF higher than 50% has been associated with an increased risk of venous thrombosis, even after adjusting for prior events, white blood cell count, and age [87,88]. There was no significant association between a JAK2V617F allele burden and arterial thrombosis [89]. Moreover, Nikolova did not find a statistically significant difference between JAK2 V617F mutation and the frequency of thrombotic events [90].

Recently, Pasquer et al. included the presence of the JAK2V617F mutation in the Venous Risk Score [91]. According to a study by Guglielmelli, the risk of venous thrombosis at any time was significantly higher in patients with the JAK2V617F mutation compared to those with CALR mutations [88]. Conversely, Zhang et al., in their study of 1537 patients with JAK2V617F, found that reducing the JAK2V617F allele burden could help prevent thrombosis, as well as control hematocrit levels and mitigate cardiovascular risk factors [67].

CALR-mutated patients had a significantly lower arterial or venous thrombotic risk than patients with a JAK2 mutation (HR: 0.346; 95% CI: 0.172, 0.699) in Wille et al. study [56]. Additionally, a study by Narlı et al. involving Turkish patients with essential thrombocythemia found that patients with mutant CALR were younger at diagnosis, had higher platelet counts, and lower hemoglobin levels than those with the mutant JAK2V617F [92]. No differences were observed in the incidence of thrombotic events between patients with type 1 and type 2 CALR mutations [92]. On the other hand, Pérez Encinas et al.’ multicentric study concluded that CALR mutation type has prognostic value for the stratification of thrombotic risk in ET patients finding that CALR-type 2 mutation was a statistically significant protective factor for thrombosis [93].

Recently, Morath et al. reported the results of a study involving MF patients. They found JAK2-V617F was identified in 55% patients, CALR in 32% patients, and 5% patients with an MPL variant [25]. JAK2-V617F mutation was associated with an increased risk of venous thrombosis (OR 2.6, 95% CI 1.01–7.16). In contrast, the DNMT3A mutation served an independent predictor of arterial thrombotic events (OR 5.40, 95% CI 1.30–22.42) [25]. Interestingly, a high JAK2-V617F variant allele frequency of greater than 50% did not correlate with an increased risk of thrombosis in their MF cohort [25].

The relationship between JAK2 and TET2 mutations in MPN patients and their association with thrombosis has been investigated. Wang et al. concluded that TET2 mutation may be more useful for predicting thrombosis in ET patients compared to PV [94]. Furthermore, ET patients with TET2 mutations were older and exhibited different patterns in coagulation markers compared to those without the mutation [94]. Significantly higher D-Dimer, lower antithrombin III), and a higher percentage of D-Dimer and fibrin degradation products were found in ET patients with TET2 mutations [94]. Ortmann et al. found that PV patients with an initial JAK2 mutation had a higher risk of thrombosis, while those with an initial TET2 mutation had a more indolent disease course [24].

Mroczkowska-Bękarciak et al. recently introduced NGS panel testing, a tool that can simultaneously detect mutations in multiple genes and quantify mutation burden, which serves as a basis for personalized prognosis prediction. However, there are limitations associated with this method, such as the requirement for highly trained staff and robust bioinformatics support [95].

The JAK2V617F mutation is the primary genetic alteration associated with thrombosis in MPN patients. A comprehensive understanding of the entire mutational landscape is crucial for accurately assessing thrombotic risk, tailoring therapeutic strategies, and implementing effective monitoring protocols.

### 4.5. Risk Scores

Based on the identified risk factors, several risk scores have been developed to stratify thrombotic risk in MPN patients (Table 1).

Initially, the conventional approach considered only age over 60 years and a history of thrombosis [49]. Subsequent risk models incorporated additional variables, such as genetic data. The IPSET score included leukocyte count greater than 11 × 10^9^/L [96], the IPSET-thrombosis model considered the presence of the JAK2V617F mutation and of the cardiovascular risk factors [97], while the revised IPSET-thrombosis model included only the presence of the JAK2V617F mutation [99]. All these risk scores were reported in ET setting. Haider et al. performed the external validation of revised IPSET-thrombosis in a ET cohort [103]. Guglielmelli et al. studied IPSET score in a cohort of pre-PMF patients, and demonstrated that IPSET score was superior to both the conventional and the revised IPSET in predicting thrombosis [98].

Mancuso et al. reported that the high-risk patients were the most represented (70.8%, 70.3%, and 60.9%, respectively) either using the traditional system, the IPSET thrombosis, or the revised IPSET thrombosis score for the assessment of the thrombotic risk [104]. Interestingly, Saelue et al. found that the conventional model had optimal calibration and a good discrimination (C-index, 0.67; 95%CI: 0.55–0.79), compared to the IPSET thrombosis (C-index 0.33; 95%CI: 0.20–0.49) and revised IPSET thrombosis (C-index 0.31; 95%CI: 0.18–0.44) in a Thai ET cohort [105].

Alvarez-Larrán et al. performed the first evaluation of IPSET-thrombosis and revised IPSET-thrombosis scores as thrombosis predictors in a ET large cohort with prospective follow-up. Both risk scales identified patients at increased risk of arterial thrombosis but failed to predict venous thrombosis [106].

Pasquer et al. recently proposed different risk scores for arterial and venous thrombotic events risk assessment in ET, PV, MF, and unclassified cases. The Arterial Risk Score included data on the cardiovascular risk factors (including male sex, tobacco, hypertension, diabetes, or hypercholesterolemia),TET2 or DNMT3A mutation, age at diagnosis of more than 60 years, and arterial thrombosis prior to or at diagnosis, while the Venous Risk Score included only venous thrombosis history prior to or at diagnosis, and JAK2 V617F mutation [91].

Elsayed et al. highlighted the limitations and utility of artificial intelligence (AI) and machine learning (ML) technologies in MPN. These models may review a large volume of data, identifying patterns and trends more easily. However, they proposed these tools as complements to the standard diagnostic criteria and clinical judgment [107].

Verstovsek et al. proposed a ML model (PV-AIM) for ET patients treated with hydroxyurea and validated it using an independent dataset. They included demographic, clinical, and laboratory data, highlighting the importance of the lymphocyte percentage, neutrophil percentage, and RDW as key markers of TE risk [102].

After discovering the independent risk factors of thrombosis, Gu et al., assigned coefficient-weighted scores to each risk factor, and developed a multiple factor-based prognostic score system of thrombosis (MFPS-PV). Age greater than 60 years, the presence of the cardiovascular risk factors, mutation for thrombosis (DNMT3A, ASXL1, or BCOR/BCORL1), and previous thrombosis made part of this score [61]. The authors suggested that MFPS-PV can identify high-risk patients ignored by the conventional model and non–high-risk patients over-evaluated by the conventional model [61].

Abu-Zeinah et al. developed an ML algorithm-based clinical decision support system that predicts the incidence of thrombosis in the following 3 months in PV patients, starting from the most significant 60 clinicopathologic variables. They used a random survival forest (RSF) model, a point beeing allocated for the presence of each of the five variables: age, old thrombosis, recent thrombosis, leukocytosis, and recent diagnosis [100].

Krichevsky et al. used ML to review high-volume clinical data to predict near-term thrombosis risk in PV. The prospective validation demonstrated the model’s generalizability. They used intrinsic factors (age, blood type), disease events (time since diagnostics/thrombosis), and short-term changes (body mass index) to tailor risk mitigation strategies [101].

Khosla et al. reported a ML algorithm to predict thrombosis in ICU patients, diagnosed with MPN. The top ten variables included in the deep learning model showed that the top ten variables associated with venous thrombosis were oxygen saturation, length of stay, blood urea nitrogen, other insurance type, anion gap, respiratory rate, bicarbonate, pressure of oxygen, pH value, and race other than white [108].

## 5. Treatment Insights

### 5.1. Prevention of the Thrombotic Complications in Myeloproliferative Disorders

Much of the current treatment recommendations in MPN setting focus on preventing thrombotic complications.

Careful risk stratification in these cases is essential to establish the appropriate management in these settings. In low-risk patients, low-dose aspirin and phlebotomy are recommended to maintain a hematocrit level lower than 45%, while in high-risk patients, cytoreduction is usually started in addition to aspirin and phlebotomy [108,109]. According to the ECLAP study, aspirin is considered a successful treatment for thrombosis prevention in PV disease [110]. There are also some reports recommending aspirin twice a day for patients at high risk of arterial thrombosis [111,112]. However, this approach is not currently endorsed by guidelines, and the competing risk of increased bleeding must be considered and cautiously balanced. In the presence of extreme thrombocytosis, the use of aspirin may exacerbate bleeding.

A selective JAK inhibitor, Ruxolitinib, has shown significant effectiveness in controlling hematocrit and symptoms in patients with myeloproliferative disorders who are intolerant or have developed resistance to hydroxyurea [113,114]. On the other hand, a recent meta-analysis of clinical trials using Ruxolitinib failed to demonstrate that its use reduces thrombosis in polycythemia vera patients compared to the previously available therapy [115]. Interferon-alpha (IFNα), an immune-modulating cytoreductive agent, has opened a new perspective for treating patients with myeloproliferative disorders [116,117]. Interferon α-2 has been evaluated successfully in the treatment of these patients [118]. However, the exact mechanism of its selective suppression of the myeloproliferative clone is still unknown [119].

Studies demonstrated that MPN arterial thrombosis was associated with increased SSC risk [5,10,11,12]. Screening for SSC in MPN patients with arterial thrombosis may be valuable in this clinical context [H].

### 5.2. Treatment of Acute Thrombotic Events in Myeloproliferative Disorders

Traditionally, the treatment of acute thrombotic events of large arterial trunks is based on anticoagulants in the acute phase, followed by aspirin for secondary prevention. However, in MPN, the efficacy of anticoagulants in the treatment of thrombotic events in large arteries is still under debate.

Baysal et al. recently conducted the largest analysis of MNP patients to date, comparing the efficacy and safety of direct oral anticoagulants (DOACs) with vitamin K antagonists (VKAs). They suggest that DOACs are promising alternatives to VKAs for managing thrombotic risk in MPNs, with an acceptable level of toxicity [120]. The thrombotic and bleeding risks were similar between the two groups, with *p*-values of 0.708 and 0.158, respectively. The incidence of thrombosis was 1.1% in the VKA group and 1.9% in the DOAC group. Additionally, the incidence of major bleeding was 0.6% for the VKA group and 1.6% for the DOAC group [120]. Similarly, Lee et al. reported that DOACs are both effective and safe in the MPN context [121]. Previously, Barbui et al. found that DOACs and VKAs exhibit a substantially similar risk–benefit profile for the prophylaxis of venous thromboembolism in MPN patients [122].

Considering that the aortic thrombi described in patients with these hematological disorders are ‘white thrombi’, consisting of primarily aggregated platelets, it is intuitive to treat patients with aspirin, which is a cyclooxygenase (COX) inhibitor [17]. Aspirin disrupts thromboxan A2 production by inhibiting COX-1 of the arachidonic acid cascade, and it exhibits antiplatelet action [17]. Indeed, the efficacy of aspirin in the prevention and treatment of thrombotic events in the microcirculation is well demonstrated in myeloproliferative neoplasms; however, its effectiveness in the primary prevention or treatment of large artery thrombosis is still under debate [123]. Platelet aggregation inhibitor combined with cytoreductive treatment appears to be sufficient [123]. Indeed, cytoreductive treatment is essential to reduce the platelet count below 500,000/mm^3^, relying mainly on hydroxyurea, the only agent with proven efficacy in preventing thrombotic events, but with a possible leukemogenic risk. Anagrelide has no leukemogenic risk, but is considered a second-line treatment when hydroxyurea becomes ineffective or poorly tolerated [123].

Although rare, medical treatment of extensive artery thrombosis has previously been reported. Fang et al. presented a case in which a large aortic thrombus associated with ET was medically treated with aspirin 325 mg daily, combined with cytoreductive therapy with hydroxyurea [17]. They adopted this management strategy due to the ET diagnosis and the histology of the aortic thrombus, which was a white thrombus consisting primarily of aggregated platelets with a minimal fibrin network and almost no entrapped erythrocytes. This case was the first reported of an extensive intra-aortic thrombosis associated with ET that has been successfully managed with medical therapy alone. In this case, symptoms disappeared within two days (similar to our case), and the disappearance of the thrombus was confirmed three weeks later by CT scan. In our case, we observed the reduction in the number and dimensions of the aortic thrombi at the repeated TOE and CT-scan on the 7th day of medical treatment. However, our patient received anticoagulant therapy with unfractionated heparin alongside aspirin and cytoreduction therapy. Our patient’s favorable and uneventful evolution demonstrates that pharmacological treatment is effective and can prevent necrosis in the organ. This implies that medical therapy can be used as the primary treatment for this condition in the future.

On the other hand, surgical treatment with thrombus removal prevents necrosis in the organ due to a potential embolism in the peripheral arteries. Although the surgical removal of large vessel thrombus was performed exceptionally, there are reported cases of successful aortic thrombectomy in ET patients and intra-aortic thrombus [14,16]. The removed thrombus consisted of platelets, and it was termed ‘white clot’ due to the characteristic thrombocythaemia [14]. However, aortic thrombectomy is a high-risk procedure. Ehrenfeld et al. reported the case of a patient who died of shock after an aortic thrombectomy for thrombocytosis [124]. In our case, surgical thrombectomy was not feasible, considering the multiple thrombi located in both large arteries and veins. Finally, emergency endovascular intervention does not seem appropriate, with a mortality rate of up to 52%, due to acute renal failure, hemorrhagic syndrome or ischemic colitis [125].

## 6. Conclusions and Future Directions

There remain critical unresolved issues concerning the pathophysiology of thrombosis within the MPN context of myeloproliferative neoplasms (MPNs), which significantly impact intervention strategies. Advancements in next-generation sequencing (NGS) panel testing are emerging as vital components of the solution [95]. However, the requirement for highly trained personnel and robust bioinformatics support poses substantial challenges in clinical practice. By investigating the molecular and functional changes that the JAK2V617F mutation induces in ECs, we may develop new treatment strategies applicable not only in MPN but also in conditions such as atherosclerosis. This is especially important because ECs and hematopoietic cells originate from the same precursor cells, and somatic mutations can be transmitted to ECs through various pathways.

Nonetheless, effective risk assessment and stratification are essential, as preventing thrombotic events is a crucial focus in MPN management. Currently, the absence of a definitive risk scale hampers progress in this area. However, leveraging clinical and genetic variables can significantly enhance the information available to clinicians. The PROSPERO study aims to identify and validate new variables that will help in developing precise and integrated prediction models, although it focuses solely on high-risk PV patients 72]. The integration of machine learning algorithms has the potential to revolutionize our approach to addressing complex challenges in atherogenesis and thrombogenesis.

Another challenge is related to the bidirectional relationship between MPN thrombosis and SSC. Further research is essential to determine whether MPN patients are predisposed to SSC, regardless of their use of cytoreductive therapy. Additionally, it is important to explore whether the duration and cumulative dose of cytoreductive therapy affect the incidence of SSC, confirm the potential protective IFN role, and identify other factors that may contribute to the development of SSC.

Balancing the risks of thrombosis and bleeding in MPN patients remains a significant challenge in treatment. Direct oral anticoagulants (DOACs) appear to offer a potential solution [120,121,122]; however, additional prospective studies are necessary to confirm their effectiveness. Another promising area of research is the potential of α-ketoglutarate (α-KG) supplementation, which may help reduce platelet hyperreactivity in MPN patients [35]. More studies are urgently needed to investigate this possibility. Additionally, effectively managing modifiable risk factors such as diabetes, obesity, hyperlipidemia, and hypertension is essential but demands comprehensive strategies that go beyond specific MPN treatments. Future therapies should focus on targeting the complex mechanisms involved in both atherogenesis and thrombogenesis, including new cytoreductive drugs targeting the somatic mutations, such as interferon and Jak2 inhibitors, and anti-inflammatory drugs for primary and secondary prevention of thrombosis.

## Figures and Tables

**Figure 1 biomedicines-13-02543-f001:**
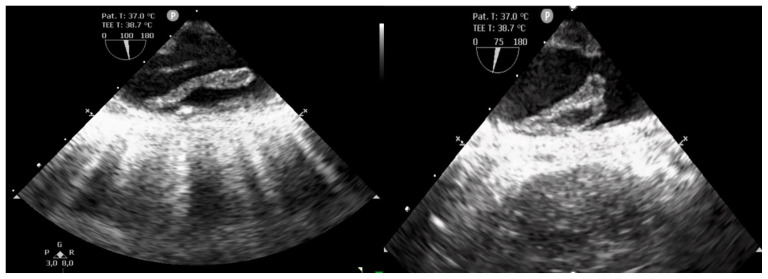
Transesophageal echocardiography showing multiple thrombi of various forms, highly mobile, attached to the wall of the descending thoracic aorta.

**Figure 2 biomedicines-13-02543-f002:**
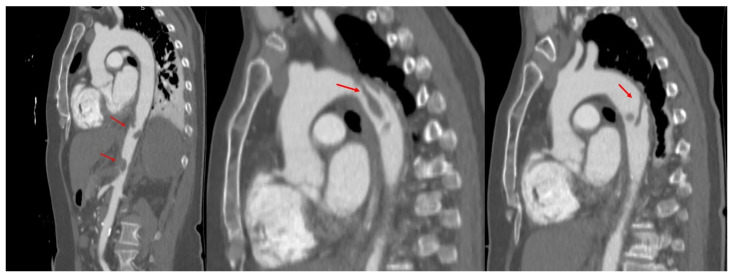
Computed tomography of the aorta, oblique sagittal images in arterial time, showing the presence of multiple thrombi (red arrows) attached to the wall of both descending, and abdominal aorta.

**Figure 3 biomedicines-13-02543-f003:**
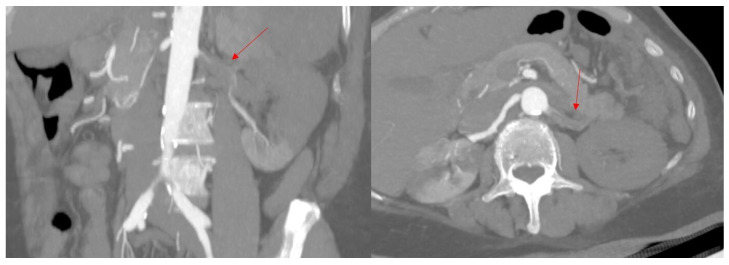
Computed tomography showing almost complete thrombosis of the left renal artery; (**left**): coronal oblique image in arterial time, (**right**): axial oblique image in arterial time. Red arrows show the left renal artery.

**Figure 4 biomedicines-13-02543-f004:**
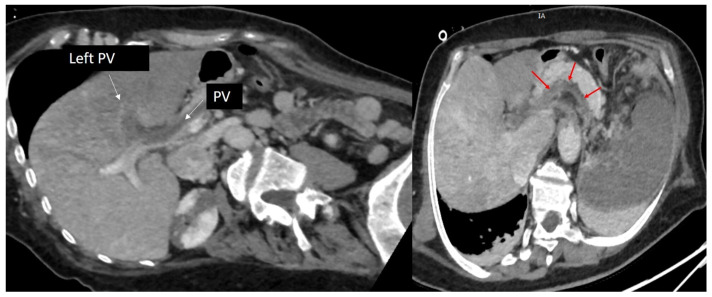
Computed tomography, oblique axial images in portal time, showing a completely obstructive thrombosis of the left portal vein extended to the portal vein trunk (red arrows). PV: portal vein.

**Figure 5 biomedicines-13-02543-f005:**
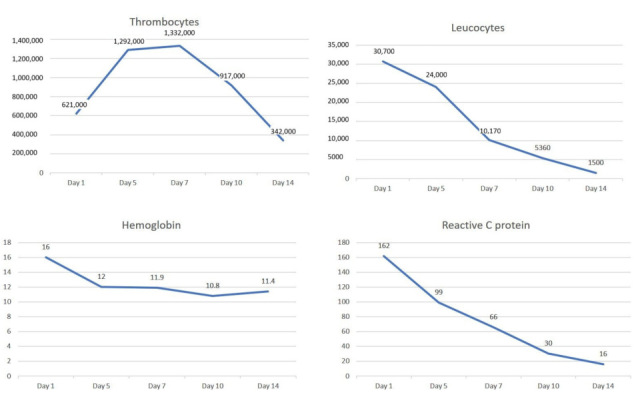
The dynamic evolution of the thrombocytes, leucocytes, hemoglobin and reactive C protein during 2 weeks of medical treatment consisting of anticoagulant, antiplatelet (aspirin) and hydroxyurea.

**Figure 6 biomedicines-13-02543-f006:**
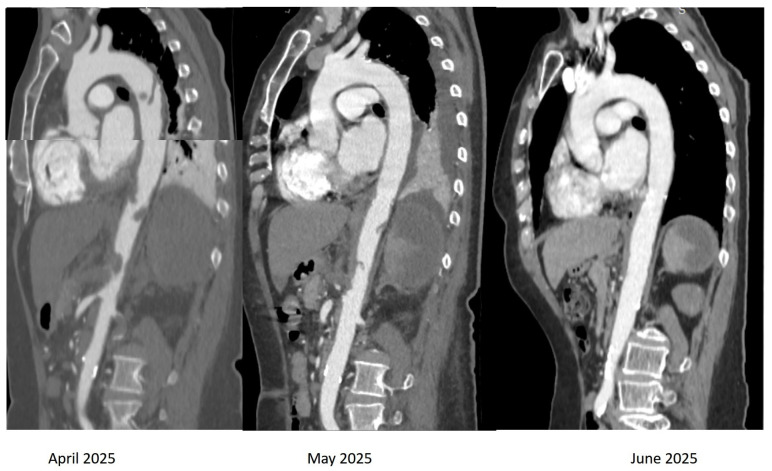
Computed tomography (CT) of the aorta, oblique sagittal images in arterial time, showing the dynamic evolution of the multiple thrombi attached to the wall of both the descending and abdominal aorta. We observe the disappearance of aortic thrombi at 3-month CT follow-up.

**Figure 7 biomedicines-13-02543-f007:**
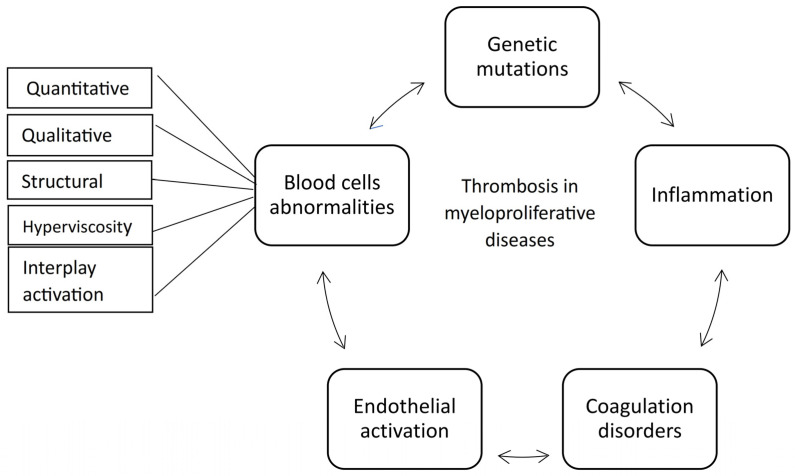
The main pathophysiological mechanisms of thrombosis in myeloproliferative neoplasms.

**Table 1 biomedicines-13-02543-t001:** The relevant risk scores for thrombotic events in myeloproliferative diseases.

Score	MPN	Variables	Categories	Reference
Conventional approach	ET	age > 60 years history of thrombosis	Low risk (no risk factors) High risk	Barbui et al. (2011) [49]
IPSET	ET	age ≥ 60 years (2 points) history of thrombosis (1 point) leukocyte count ≥ 11 × 10^9^/L. (1 point)	Low (points 0) Intermediate (points 1–2) High (points 3–4)	Passamonti et al. (2012) [96]
IPSET-thrombosis	ET	age > 60 years (1 point) thrombosis history (2 points) cardiovascular risk factors (1 point) JAK2V617F (2 points) -	Low-risk < 2 points; Intermediate risk 2 points High-risk > 2 points	Barbui et al. (2012) [97]
	pre-PMF			Guglielmelli et al. (2020) [98]
Revised IPSET-thrombosis	ET	age > 60 years history of thrombosis JAK2V617F	Very low (age ≤ 60 years, JAK2 wild type, no prior thrombosis) Low (age ≤ 60 years, JAK2 V617F+, no prior thrombosis) Intermediate (age > 60 years, JAK2 wild type, no prior thrombosis) High (age > 60 years and JAK2 V617F+, or prior thrombosis history regardless of other factors)	Barbui et al. (2015) [99]
Machine learning	PV	age ≥ 60 years—1 point; prior thrombosis—1 point; leukocyte count ≥ 12 × 10 ^9^/L—1 point peri-diagnosis thrombosis (<2 years from diagnosis)—1 point peri-thrombosis (<2 years from last thrombosis)—1 point	High-risk (score ≥ 2) Intermediate-risk (score = 1) Low-risk (score = 0)	Abu-Zeinah et al. (2021) [100]
Machine learning	PV	intrinsic factors (age, blood type), disease events (time since diagnosis/thrombosis), and short-term changes (body mass index)	-	Krichevsky et al. (2023) [101]
Machine learning (PV-AIM)	ET	included demographic, clinical, and laboratory data		Verstovsek et al. (2023) [102]
MFPS-PV	PV	age ≥ 60 years cardiovascular risk factors mutation for thrombosis (DNMT3A, ASXL1, or BCOR/BCORL1) previous thrombosis	-	Gu et al. (2023) [61]
Arterial Risk Score (ARTS)	ET PV MF MDS unclassified	cardiovascular risk factors (male sex, tobacco use, hypertension, diabetes, or hypercholesterolemia), 1 point; *TET2* or *DNMT3A* mutation, 1 point; age at diagnosis > 60 years, 2 points; arterial thrombosis prior to or at diagnosis, 2 points.	Low risk (score 0–1) High risk (score 2–6).	Pasquer et. al. (2024) [91]
Venous Risk Score	ET PV MF MDS unclassified	venous thrombosis prior to or at diagnosis, 1 point; *JAK2 V617F* mutation, 1 point.	Low risk (score 0) High risk (score 1–2).	Pasquer et al. (2024) [91]

Abbreviations: ET, essential thrombocythemia; IPSET-thrombosis—International Prognostic Score of thrombosis; pre-PMF, pre-fibrotic myelofibrosis; PV, polycythemia vera; MDS, myelodisplazic syndrome; MF, myelofibrosis; MPN, myeloproliferative neoplasms.

## Data Availability

No new data were created or analyzed in this study. Data sharing is not applicable to this article.

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
