# Peer review of "Clinical and Molecular Insights of Arterial and Venous Thrombosis in Myeloproliferative Diseases—Case-Based Narrative Review"

_biomedicines, 2025, doi:10.3390/biomedicines13102543_

Round 1
Reviewer 1 Report
Comments and Suggestions for Authors
This is a comprehensive review on thrombosis in MPN, faclitated by a case report to put the review in context of direct patient care. Thrombosis is sometimes an underappreciated clinical aspect of MPN so it is important, as this paper does, to shed more emphasis on thrombosis.
The review does a very nice job at comprehensively reviewing the literature on multiple aspects of thrombosis in MPN. The manuscript is of interest to clinicians and scientists alike, as it contains clinical pearls as well as emphasizes mechanistic unknowns of what mediates thrombosis in MPN.
I only have a few minor (mostly grammatical comments)
- Right after 2. Case Presentation - it appears there is some retained instruction language in the document that is not part of the manuscript
- Few misspelled words - leukocitosis, absolut
- The patient is referred to as MPN-U, however I noticed her hgb was 16.0 at presentation. Why would this patient not be considered PV?
Reviewer 2 Report
Comments and Suggestions for Authors
Drăgan et al reported a rare case of MPN associated thrombosis, and based on this case, reviewed clinical and genetic insights into thrombosis associated with MPN. Various therapeutic options for these conditions were also discussed. This review is very comprehensive and pleasant to read.
Comments:
- A division of the section 3 into subsections, e.g. genetics, blood cells, et al., would be helpful for the readers to follow.
- Lines 222-251, the descriptions of JAK2 mutations associate with MPN thrombosis seemed to overlap with those at lines 533-552. Consider shorten one of its appearances.
- The authors exhaustly listed recent studies on the associations between genetic and cellular factors with MPN-thrombosis, however, it seems to me that it lacks sentences to summarize the evidence from different studies on the thrombosis predictivity for each possible risk factors. For some factors consistent observations were found, while contradictory observations were made for other factors.
Minor:
- The paragraph between lines 97-102 should be deleted.
- In several places there seems to be unsuccessfully inserted references. For example, Line 269, “[Giaccherini]”; line 302 “[Liu]”. Please double check.
Reviewer 3 Report
Comments and Suggestions for Authors
For the introduction part, BCR-ABL1-negative MPN is a more definitive term than Philadelphia chromosome-positive MPN.
The beginning of the Case presentation part “The Materials and Methods should be described with sufficient details to allow others to replicate and build on the published results. Please note that the publication of your manuscript implicates that you must make all materials, data, computer code, and protocols associated with the publication available to readers. Please disclose at the submission stage any restrictions on the availability of materials or information. New methods and protocols should be described in detail while well-established methods can be briefly described and appropriately cited” I think this is a mistake and should be deleted.
For the pathophysiological mechanisms of thrombosis in MPN. It was also demonstrated that endothelial cells harboring the JAK2V617F mutation can also enhance thrombosis. This article should be read and cited (DOI: 10.4274/tjh.galenos.2024.2024.0161). It can be considered as the further development of the work described in reference 37.
Also, there is an increase in secondary cancers in patients with MPM who developed arterial thrombosis. It was shown in two real-world studies (https://doi.org/10.1038/s41408-024-01052-4) and (DOI: 10.4274/tjh.galenos.2024.2024.0199). These should be read and cited.
Another aspect is enhancing the value of the treatment and thromboprophylaxis in MPN. There is a growing body of studies on the DOAC in MPN. There are two studies with the largest number of patients to date on the use of DOACs in MPN, and these studies reported that DOACs are effective and safe. These articles should be added to the references. (https://doi.org/10.1038/s41375-021-01279-1) and (doi: 10.1007/s11239-024-03043-5).
The conclusion and future directions part should also be rewritten. For example, α-ketoglutarate (α-KG) supplementation was not mentioned in the body of the manuscript, so why should it play a role in the future? The PROSPERO trial solely investigates high-risk PV patients; therefore, it will not be the answer for all MPN types. In addition, appropriate references should be used and cited for the mentioned articles or studies.
